Low-cost intelligent surveillance system based on fast CNN

Sabri Zaid Saeb 1 2
Li Zhiyong zhiyong.li@hnu.edu.cn 1
1 College of Computer Science and Electronic Engineering, Hunan University , Changsha , Hunan , China
2 Department of Computer Science & Information Systems, Al-Mansour University College , Baghdad , Al-Andalus Square , Iraq
Bernardi Mario Luca
Electronic publication date: 2021 Feb 25
Publication date: 2021
Volume: 7
Electronic Location ID: e402
Received 2020 Apr 28; Accepted 2021 Jan 29
Copyright: ©2021 Sabri and Li
Copyright year: 2021
Copyright holder: Sabri and Li
License: This is an open access article distributed under the terms of the Creative Commons Attribution License, which permits unrestricted use, distribution, reproduction and adaptation in any medium and for any purpose provided that it is properly attributed. For attribution, the original author(s), title, publication source (PeerJ Computer Science) and either DOI or URL of the article must be cited.
License URL: https://creativecommons.org/licenses/by/4.0/

Keywords: Surveillance system, Pattern recognition, Conversional neural network, Saliency, Raspberry PI, Raspbian OS, Neural network, Computer vision, Low-cost, Object tracking

Funding: National Natural Science Foundation of China 61672215 61976086 National Key R&D Program of China 2018YFB1308604 Hunan Science and Technology Innovation Project 2017XK2102 This work was supported by the National Natural Science Foundation of China (No.61672215, 61976086), National Key R&D Program of China (No. 2018YFB1308604), and Hunan Science and Technology Innovation Project (No. 2017XK2102). The funders had no role in study design, data collection and analysis, decision to publish, or preparation of the manuscript.

==============================
Smart surveillance systems are used to monitor specific areas, such as homes, buildings, and borders, and these systems can effectively detect any threats. In this work, we investigate the design of low-cost multiunit surveillance systems that can control numerous surveillance cameras to track multiple objects (i.e., people, cars, and guns) and promptly detect human activity in real time using low computational systems, such as compact or single board computers. Deep learning techniques are employed to detect certain objects to surveil homes/buildings and recognize suspicious and vital events to ensure that the system can alarm officers of relevant events, such as stranger intrusions, the presence of guns, suspicious movements, and identified fugitives. The proposed model is tested on two computational systems, specifically, a single board computer (Raspberry Pi) with the Raspbian OS and a compact computer (Intel NUC) with the Windows OS. In both systems, we employ components, such as a camera to stream real-time video and an ultrasonic sensor to alarm personnel of threats when movement is detected in restricted areas or near walls. The system program is coded in Python, and a convolutional neural network (CNN) is used to perform recognition. The program is optimized by using a foreground object detection algorithm to improve recognition in terms of both accuracy and speed. The saliency algorithm is used to slice certain required objects from scenes, such as humans, cars, and airplanes. In this regard, two saliency algorithms, based on local and global patch saliency detection are considered. We develop a system that combines two saliency approaches and recognizes the features extracted using these saliency techniques with a conventional neural network. The field results demonstrate a significant improvement in detection, ranging between 34% and 99.9% for different situations. The low percentage is related to the presence of unclear objects or activities that are different from those involving humans. However, even in the case of low accuracy, recognition and threat identification are performed with an accuracy of 100% in approximately 0.7 s, even when using computer systems with relatively weak hardware specifications, such as a single board computer (Raspberry Pi). These results prove that the proposed system can be practically used to design a low-cost and intelligent security and tracking system.

Introduction

Most traditional video surveillance systems use surveillance cameras connected to monitors and screens. However, in recent times, the need to detect and classify normal or abnormal events to suitably assess a given situation and to adopt security measures has emerged. Typically, in standard surveillance systems that involve a large number of surveillance cameras covering a large area, certain operators must continuously check the real-time footage recorded by the cameras (Fig. 1). In the event of an undesirable incident, the operators must alert security or the police. In certain surveillance camera systems, the monitors display the video stream from a single camera. However, in most cases, a single monitor displays multiple streams from several cameras, such as 4, 8, or 16 cameras, in a sequential or simultaneous manner (Aldasouqi & Hassan, 2010). Furthermore, practically, the operators cannot monitor all the screens all the time; instead, the camera output is recorded by using video recorders, such as digital video recorder (DVR) or network video recorder(NVR). If an incident occurs, video footage is used as evidence. One drawback of this strategy is that the operators cannot address the incidents or prevent any related damage in real time as the recordings can only be viewed at a later time. Moreover, considerable time is required to find the relevant section of the recording as a suspect is often at the scene long before an incident occurs, and the recording may correspond to multiple cameras. Consequently, it is necessary to develop a method or technique that can instantly analyze and detect threats based on the detection of humans and their activities (Salahat et al., 2013; Troscianko et al., 2004).

In the last decade, modern video surveillance systems have attracted increasing interest with several studies focusing on automated video surveillance systems, which involve a network of surveillance cameras with sensors that can monitor human and nonhuman objects in a specific environment. Pattern recognition can be used to find specific arrangements of features or data, which usually yield details regarding a presented system or data set. In a technical context, a pattern can involve repeating sequences of data with time, and patterns can be utilized to predict trends and specific featural configurations in images to recognize objects. Many recognition approaches involving the use of the support vector machine (SVM) (Junoh et al., 2012), artificial neural network (ANN) (Petrosino & Maddalena, 2012), deep learning (Wang et al., 2019), and other rule-based classification systems have been developed. Performing classification using an ANN is a supervised practical strategy that has achieved satisfactory results in many classification tasks. The SVM requires fewer computational requirements than the ANN; however, the SVM provides lower recognition accuracy than the ANN. In recent years, networks have played a significant role in a wide range of applications, and they have been applied to surveillance systems. In recent years, as the amounts of unstructured and structured data have increased to big data levels, researchers have developed deep learning systems that are basically neural networks with several layers. Deep learning allows one to capture and mine larger amounts of data, including unstructured data. This approach can be used to model complicated relationships between inputs and outputs or to find patterns. However, the associated accuracy and classification efficiency are generally low (Liu & An, 2020). Many strategies have been developed to increase the recognition accuracy. In this work, we discuss the accuracy gains from adopting certain saliency methods to improve the recognition and detection of an object and isolate it from a scene.

Figure 1 A typical control room pertaining to traditional surveillance systems.

©123rf.com.

The performance efficiency of existing surveillance systems is highly dependent on the activity of human operators who are responsible for monitoring the camera footage (Sedky, Moniri & Chibelushi, 2005). In general, most medium and large surveillance systems involve numerous screens (approximately 50 or more) that display the streams captured by numerous cameras. As the number of simultaneous video streams to be viewed increases, the work of surveillance operators becomes considerably challenging and fatiguing. Practically, after twenty minutes of continuous work, the attention of the operators is expected to degrade considerably. In general, the operators check for the absence or presence of objects (i.e., people and vehicles) in surveillance areas and ensure that the maximum capacity of a place remains intact, such as by ensuring that no unauthorized people are present in restricted areas and no objects are present in unexpected places. The failures of such systems in alarming authorities can be attributed to the limitations of manual processing. Generally, most traditional methods used to obtain evidence depend heavily on the records of the security camera systems in or near accident sites. Practically, when an incident occurs in a vast space or considerable time has elapsed since its occurrence, it is difficult to find any valuable evidence pertaining to the perpetrators from the large number of surveillance videos, which hinders the resolution of the cases. Thus, to minimize the mental burden of the operators and enhance their attention spans, it is desirable that an automated system that can reliably alert an operator of the presence of target objects (e.g., a human) or the occurrence of an anomalous event be developed.

Pattern recognition, which is widely used in many recognition applications, can be performed to find arrangements of features or data, and this technique can be applied in the surveillance domain. Several recognition approaches involving the support vector machine, artificial neural networks, decision trees, and other rule-based classification systems have been proposed. Machine learning typically uses two types of approaches, namely, supervised and unsupervised learning. Using these approaches, especially supervised learning, we can train a model with known input and output data to ensure that it can estimate any future output. Moreover, in some existing systems, an artificial immune system (AIS)-inspired framework, where the AIS is a computational paradigm that is a part of the computational intelligence family and is inspired by the biological immune system that can reliably identify unknown patterns within sequences of input images, has been utilized to achieve real-time vision analysis designed for surveillance applications (Cserey, Porod & Roska, 2004).

Literature Survey

The field of video surveillance is very wide. Active research is ongoing in subjects, such as automatic thread detection and alarms, large-scale video surveillance systems, face recognition and license plate recognition systems, and human behavior analysis (Mabrouk & Zagrouba, 2018). Intelligent video surveillance (Singh & Kankanhalli, 2009) is of significant interest in industry applications because of the increased requirement to decrease the time it takes to analyze large-scale video data. Relating to the terminology, Elliott Elliott (2010) recently described an intelligent video system (termed IVS) as “any kind of video surveillance method that makes use of technology to automatically manipulate process and/or achieved actions, detection, alarming and stored video images without human intervention.” Academic and industry studies are focused on developing key technologies for designing powerful intelligent surveillance systems along with low-cost computing hardware; and the applications include object tracking (Khan & Gu, 2010; Avidan, 2007), pedestrian detection (Dalal & Triggs, 2005), gait analysis (Wang, 2006), vehicle recognition (Wang & Lee, 2007), privacy protection (Yu et al., 2008), face and iris recognition (Park & Jain, 2010), video summarization (Cong, Yuan & Luo, 2012) and crowd counting (Cong et al., 2009). Nguyen (Nguyen et al., 2015) described the implementation and design of an intelligent low-cost monitoring system using a Raspberry Pi and a motion detection algorithm programmed in Python as a traditional programming environment. Additionally, the system utilizes the motion detection algorithm to considerably reduce storage usage and save expense costs. The motion detection algorithm is executed on a Raspberry Pi that enables live streaming cameras together with motion detection. The real-time video camera can be viewed from almost any web browser, even by mobile devices. Sabri et al. (2018) present a real-time intruder monitoring system based on a Raspberry Pi to deploy a surveillance system that is effective in remote and scattered places, such as universities. The system hardware consists of a Raspberry Pi, long-distance sensors, cameras, a wireless module and alert circuitry; and the detection algorithm is designed in Python. This system is a novel cost-effective solution with good flexibility and improvement needed for monitoring pervasive remote locations. The results show that the system has high reliability for smooth working while using web applications; in addition, it is cost-effective. Therefore, it can be integrated as several units to catch and concisely monitor remote and scattered areas. Their system can also be controlled by a remote user geographically or sparsely far from any networked workstation. The recognition results prove that the system efficiently recognized intruders and provided alerts when detecting intruders at distances between one to three meters from the system camera. The recognition accuracy is between 83% and 95% and the reliable warning alert is in the range of 86–97%. Turchini et al. (2018) proposes an object tracking system that was merged with their lately developed abnormality detection system to provide protection and intelligence for critical regions.

In recent years, many studies have focused on using artificial intelligence for intelligence surveillance systems. These techniques involve different approaches, such as the SVM, the ANN, and the latest developed types based on deep learning techniques. However, deep neural networks are computationally challenging and memory hungry; therefore, it is difficult to run these models in low computational systems, such as single board computers (Verhelst & Moons, 2018). Several approaches have been utilized to address this problem. Many approaches have reduced the size of neural networks and maintained the accuracy, such as MobileNet, while other approaches minimize the number of parameters or the size (Véstias, 2019).

System Concepts

We designed a robust surveillance system based on the faster RCNN and enhanced it by utilizing a saliency algorithm. The following equation can be used to determine the dimensions of the activation maps (O’Shea & Nash, 2015; Aggarwal, 2018): (1) Di+2Pa−Df∕St+1;

where Di = Image dimension (input file)

• Pa = Padding

• Df = Filter dimension

• St = Stride

A CNN has a certain activation range. In this work, we used a rectified linear unit or ReLU function. Currently, the ReLU is one of the most commonly used activation functions in NNs. One of the most significant advantages of the ReLU over other activation functions is that it is unable to activate all neurons at the same time. The ReLU function transforms all the negative inputs to 0, and no neuron is activated. Consequently, the function is computationally efficient since only a few neurons are activated over time. Practically, the ReLU converges six times faster than the sigmoid and tanh activation functions. One of the disadvantages of the ReLU is that it is saturated in the negative region, which means that the gradient in that region is 0. In this case, all the weights are not updated through backpropagation (BP), and a leaky ReLU can be used to overcome this limitation. In addition, ReLU functions are not centered at zero, which means that a random and thus longer path is often adopted for the functions to reach their optimal points. In addition, a pooling layer is placed between the convolution layers. The pooling layer fundamentally minimizes the amount of computation and number of parameters in the network and controls the overfitting by progressively minimizing the spatial size of the network. Generally, two operations are performed in this layer: maximum and average pooling. In this work, we utilize the max pooling technique. Specifically, only the maximum value is obtained from the pool by using filters sliding throughout the input; and at each stride, the maximum parameter is extracted, and the remaining values are not considered. This technique practically downsamples the network. Compared with the convolution layer, this layer does not alter the network, and the depth dimension remains unchanged (Shang et al., 2016).

The output after performing max pooling can be determined as (2) Di−Df∕St+1

where

• Di = Dimension of input (image) to pooling layer

• F = Filter dimension

• St = Stride

In the fully connected layer, all the neurons are fully connected to each activation from the prior layers. These activation values can be computed via matrix multiplication and then a bias offset, which is the last phase of the CNN. The CNN is constructed using hidden layers and fully connected layers.

The RCNN (Girshick et al., 2014) extracts many parts from the presented image utilizing selective search and then investigates whether any of these boxes has an object. First, the model extracts all these regions; and for every region, a CNN is utilized to extract specific features. Finally, these features are later used to detect objects. However, the RCNN is slow because of these multiple steps included in the process. The fast RCNN (Girshick, 2015), alternatively, passes the entire image towards the convolutional net that yields regions of interest (rather than transferring the extracted areas from the image). Additionally, rather than making use of three different models (as in the RCNN), it utilizes a single model that extracts features out of the areas, classifies them into several classes, and applies bounding boxes. Each of these steps is conducted at the same time, hence making it execute quicker than the RCNN. However, the fast RCNN is not fast enough in cases where it is applied to a large dataset since it also uses selective search for region extraction. The faster RCNN (Ren et al., 2016) improves upon the fast RCNN. In the faster RCNN method, the “Selective Search” method was replaced by the Region Proposal Network (known as RPN), which is a network to present regions (Brandenburg et al., 2019) and is faster than the RCNN and fast RCNN.

To improve the recognition process and reduce features, we utilize a saliency algorithm to enhance the image maps as the algorithm maps the images to indicate the unique quality of each pixel.

In computer vision, the Saliency map is defined as an image in which every pixel in the image has unique quality. The aim of a saliency map is to make the image simpler and/or change it is representation (Daniilidis, Maragos & Paragios, 2010). The saliency detection approach is commonly used in the areas of cognition and target detection (Moosmann, Larlus & Jurie, 2006; Zhaoyu, Pingping & Changjiu, 2009; Kanan & Cottrell, 2010; Borji et al., 2019), object discovery (Frintrop, Garcá & Cremers, 2014), image segmentation (Kang et al., 2012; Yanulevskaya, 2013), visual tracking (Klein et al., 2010; Borji et al., 2012; Stalder, Grabner & Gool, 2012), etc. The saliency represents a type of image segmentation technique. The saliency map aims to simplify and adjust the image representation to a more substantial form that is faster and easier to analyze. For example, when a pixel has a considerably large gray level or different color values in a color image, the quality of the pixel can be identified easily in the saliency map.

The saliency can be local or global (Borji & Itti, 2012). In the local domain, the contrast corresponds to the saliency of the image patches in the local neighborhoods. In contrast, in the global domain, to determine the saliency of an image patch, the contrast is computed using the patch statistics along the whole image. In this work, we utilized the global saliency approach. The local patch is identical to its neighbors. However, the whole area (that is, the local domain and its surroundings) exhibits a global characteristic in a scene. If only the local saliency is considered, the areas may be reduced to a homogeneous area, which causes blank regions and impedes the realization of object-based focus (for example, a uniformly textured object could solely be salient at its edges). To overcome this limitation, in this work, global saliency is adopted, which is established by guiding an operator through the saliency measure of data. Instead of using each pixel, we calculate the possibility of each patch P(pi) across the whole scene and determine its inverse to obtain the global saliency Sg as follows (Borji & Itti, 2012; Ming-Ming et al., 2015): (3) logSgcpi=−logPpi

and (4) logSgcpi=−∑j=1n logPαij

To calculate Ppi, the coefficients α are considered to be conditionally independent. This aspect, to some extent, is performed by using a sparse coding algorithm. The description vector of each patch coefficient, that is, the initial binned histogram (100 bins), is determined from each of the patches in the scene and transformed to (Pαij) by dividing by the sum. In cases in which the patch is difficult to find in one of the features, the previous product is assigned a small value, which leads to a higher global saliency for the entire patch.

The proposed method is based upon the measurement of the saliency in each color space. After the measurement, the saliency values are merged into a final saliency map. For every color channel, first, the input image is separated into a nonoverlapping patch. Each patch is symbolized via a coefficient vector of the saliency from the index tree of patches derived from natural scenes. Subsequently, the global and local saliency are determined and combined to represent the saliency of each patch.

The saliency contrast maps are consolidated, and the output is calculated as follows: (5) Slg= ∫SppBp∣Idp

where I is the input image, and Slg denotes the final saliency map.

The local and global saliency maps can be normalized and merged as (6) Slgcpi=NSlcpioNSgcpi

where o is an integration structure (scheme, such as -, +, ∗, min, or max). The saliency values of the image patch in every channel are normalized and summed iteratively to determine the saliency of a patch in every color system. Figure 2 illustrates the global and local saliency of an image patch.

Figure 2 Illustration of the global and local saliency of an image patch (Sun & Li, 2018).

System Architecture

The objective of this work is to design a smart, low-cost surveillance system that can control surveillance units by using certain devices. The system is expected to monitor and control one or multiple surveillance cameras. The system should be able to detect certain aspects in a smart and automated manner for closed or open areas, such as regions between or outside buildings, and the surrounding areas. The proposed system is composed of a control unit, which controls all the processes, sensors (i.e., ultrasonic sensors to detect motion and distance), and a camera to output continuous video to identify the presence of humans and their activities. Figure 3 illustrates an example of the indoor surveillance system of buildings. The surveillance system includes cameras and ultrasonic sensors distributed to cover the main areas. The system first gathers data from the ultrasonic sensor model (HC-SR04) in real time. When the sensor detects movement in its range, it relays a signal to the camera located in that zone along with the computed distance. The camera recognizes the movements and assesses the threat. In case of any threat, the system alarms the security officer and informs him/her of the type of threat, such as if intruders (or even employees) are present in regions that are out of bounds. Figure 3 illustrates the detection operation.

Figure 3 Method to activate the camera on detecting motion.

The system can be scheduled to perform multiple tasks multiple times by using the recognition process to intelligently analyze moving objects and activity. For example, the system can activate all cameras during working times and recognize faces to identify employers and any authorized guests. If the system detects any face that is not authorized or suspicious, the surveillance officers are alarmed, and the system tracks the suspects. In addition, the system can alert officers if any employer enters an area at a prohibited time. The main concept is to implement a fast recognition system that can perform key recognition even when using compact and low computational units, such as a Raspberry Pi. Such a system can be implemented by utilizing a conventional neural network and computer vision technique known as the saliency algorithm.

Hardware Design

We employed two systems: a PC-based system, and a single board microcontroller system. The hardware specs of each system are described in Table 1.

Table 1 Hardware specs of each system.

PC-Based System	Single Board Microcontroller System	
Item	Specifications	Cost	Item	Specifications	Cost	
System	Intel NUC NUC7CJYH
2 GHz Intel Celeron Processor
8 GB Ram
128 GB SSD
	$139	System	Raspberry Pi 3 version B+
1.2 GHz Broadcom BCM2837CPU
1 GB of RAM	47$	
Camera	Logitech HD Webcam C615	85$	Room	32 GB SanDisk Ultra Micro SDHC	10$	
	Or, Commercial fast low cost 1080p Webcam	13$	Camera	Raspberry Pi Camera Version 2	24$	
Sensor	MaxBotix MB1043 HRLV MaxSonar Ultrasonic Range Finder	20$	Sensor	HC-SR04 ultrasonic sensor	2$	
Total Cost	244$ or 172$	Total Cost	83$	

Computers and single-board computers are used as processing units in most processing operations. Cameras are used to stream video from the desired area to process the images and recognize the presence of a person(s) within the camera’s field of view. The ultrasonic sensor is used to determine the distance.

For the single board microcontroller system, the following connections were made with the Raspberry Pi:

1. The camera is connected through the CSI camera port.

2. The ultrasonic sensor is linked to Raspberry Pi through 4 pins, where VCC is linked to pin 2, GND is linked to ground pin, echo is linked to GPIO 12, and trig is linked to GPIO 16.

The connection scheme for the proposed surveillance system based on a single board is shown in Fig. 4.

Figure 4 Connection scheme for the proposed surveillance system based on a single board.

Implementation of the System Program

The proposed system program is designed in Python. For both systems, Python version 3.7 installed on the operating system (OS: Windows 10 for the PC system and Raspbian for the Raspberry Pi) is used. The program is designed to manage the overall processes, starting from gathering information from the camera(s) (streaming video) and sensor(s) (signals). This program, which uses a CNN, is enhanced by using the saliency map algorithm. The algorithm analyzes the scenes continuously and classifies any detected motion. Subsequently, the algorithm isolates humans and detects threats based on human face recognition and human activity. Face recognition is performed to identify any suspicious person and alarm the observer. Moreover, the system analyzes human activity; and thus, officers are alarmed in the event of any suspicious activity, such as the presence of a gun. However, we designed our model based on work done by Rosebrock (2019) as a reference for designing a CNN model.

Methodology

As described previously, the system program is based on the CNN algorithm that is optimized using a computer vision technique known as the saliency algorithm. The proposed system involves the following steps:

Data collection

Approximately 3,450 images (of humans) and 10,014 images for three categories of weapons (knife, small gun, and large gun) are collected to perform the classification task by using transfer learning to reduce false positives. For humans, a total of 3,450 images were collected and divided into two categories: the training set and the testing set. The captured images of humans included various poses, perspectives and orientations. This can help the deep learning CNN learn the required objects in an efficient way. From the 3,450 images of humans, 2761 images were selected for training, and 689 were selected for testing. For weapons, 10,014 images are used to classify the three categories of weapons (knife, small gun, and large gun). Of these 8,011 images were used for training, and 2,003 images were used for testing. To repurpose a pretrained model, we fine-tune our model by training certain layers and freezing other layers.

Preprocessing images

This step includes several processes, such as augmentation (shift and flip), resizing, rotation, zooming and introducing Gaussian noise. The images were cropped to a square ratio and then resized to 800 ×800 pixels.

Optimization with Saliency Algorithm

The stepwise procedure of the proposed saliency algorithm can be described as follows:

Step 1: Image preprocessing: In this part, we first streamed live video as FPS images and later converted it to grayscale.

Step 2: Image separation: In this part, we segmented the image by using a superpixel algorithm (Achanta et al., 2012; Zhang, Malmberg & Sclaroff, 2019), which is the simple linear iterative clustering (SLIC) algorithm and is based on the typical k-means method in order to group pixels for conventional color areas. SLIC superpixels are made based on two criteria: one criterion is the spectral similarity (limited to 3 channels), and the second criterion is the spatial proximity.

Step 3: Extracting features from an image: In this stage, the input image is portioned to make it perceptually homogeneous and obtain tiny features by using two algorithms: the Boolean Map Saliency algorithm (BMS) (Zhang, Malmberg & Sclaroff, 2019) and applied LG (Local + Global).

Step 4: Create index tree: Along with superpixels, a particular index tree is generated to encode the construction information through hierarchical separation. Consequently, we first calculate the gain of every surrounding patch by obtaining the 1st- and 2nd-order reachable matrix (Peng et al., 2016).

Step 5: Recombining: In this part, we recombine all the patches and execute context-based propagation to obtain the final saliency map.

Step 6: Recognition: The CNN is applied to recognize the separated features and classify the specified objects in the saliency map.

These images are then annotated and stored in XML format. Then, this XML file is transformed into CSV format and then transformed into TF data that will be input into the deep learning framework. After the TF data have been generated, the training phase is started.

Modeling with MobileNet and Faster RCNN

Transfer learning is used to build appropriate models while reducing the time consumption. In this work, pretrained models are used to execute transfer learning. Due to the high computational costs of training complex models, it is a regular practice to import and use existing models (e.g., VGG 16 (Simonyan & Zisserman, 2015), Inception (Szegedy et al., 2014; Szegedy et al., 2015) or MobileNet (Howard et al., 2017)). In this work, we utilized MobileNet as a feature extractor.

The RPN is composed of two layers used to find the areas that can include objects in an image (feature maps). The network utilizes the ROI pooling layer to minimize and resize resource maps depending on proposals from that area. The maps make use of the new features of every area to select frames through three fully connected layers (FCLs). MobileNet has been used as a CNN that takes the layers as learning functions; hence, the original feature extraction has several layers. However, the first convolution stack structures are obtained via transfer learning through the use of MobileNet. The method includes two steps forming the current surveillance detection. The first step determines the ROIs in images. All these ROIs are considered references indicating several possible object sites that are created in the second step. Figure 5 shows the proposed model that consists of five convolutional layers and three FC layers. The faster RCNN mainly uses the last convolutional layer features to perform classification and localization. After the two convolutional layers, the outputs of the last three convolutional layers (layers three, four and five) are utilized as input data to the 3 levels of pooling of the ROI and the related normalization levels. For every RPN anchor forming a fully convolutional network, a degree is forecasted that makes it able to determine the probability that this anchor has the element of interest. Moreover, the RPN offers the acceleration and measurement coefficients for every anchor, which is a part of the peripheral regression mechanism, and thus enhances the position of the object.

Figure 5 The proposed approach of faster RCNN.

The structure consists of five convolutional layers and three fully connected layers.

As an illustration, the architecture includes two steps. First, the RPN presents a set of bounding boxes having a trusted rating related to a potential human image. The second step defines the analysis of those fully convolution architectures using MobileNet as a feature extractor after obtaining the output feature map from a pretrained model, which is MobileNet. As we used an input image with a resolution of 800 × 800 in x3 dimensions, the output feature map should be 50 × 50 × 256 dimensions. Every point in a 50 × 50 area represents an anchor. Hens, we must specify sizes and specific ratios for every anchor, which are 1282, 2562, and 5122 for the three respective sizes and 1:1, 1:2, and 2:1 for the three respective ratios, in the original image. Then, the RPN is linked to a conv layer using 3 ×3 filters, 1 padding, and 512 output channels. Then, the output is linked to two 1 ×1 convolutional layers for box regression and classification (where the classification is used to verify if the box is an object or is not). In such a case, each anchor will have 9 corresponding boxes from the original image, which means there are 50 × 50 × 9 = 22, 500 boxes in the original image. We only select 256 of these 22,500 boxes to be a minibatch that has 128 backgrounds (neg) and 128 foregrounds (pos). Simultaneously, nonmaximal suppression is implemented to ensure that there is zero overlap for those proposed regions. When the previous steps are finished, then the RPN is finished. In the second stage of the RCNN, similar to the fast RCNN, ROI pooling is utilized for these proposed areas (ROIs). Then, we flatten this layer using some fully connected layers. The last step is a softmax function for linear regression and classification to fix the boxes’ locations. Figure 6 shows the FRCNN/RPN structures

Figure 6 (A) First step of FRCNN, (B) RPN structures, in which k is the anchors number.

Results

This section describes the accuracy gain when the saliency method is applied to generate a saliency map, which is used as an input for the CNN to detect related objects. Figures 7 and 8 show the saliency maps generated from a live streamed video.

Figure 7 Saliency map results for humans (in real time streaming).

(A–B) Streamed images; (C–D) the output (saliency results) of streamed image (A) and (B) respectively.

Figure 8 Saliency map results for humans with a gun (in real time streaming).

(A) Streamed images; (B) the output (saliency results) of streamed image (A).

As shown in Figs. 7 and 8, the human body and gun are extracted from a highly detailed image that involves many objects. The results indicate that the proposed method efficiently removes foreground objects (humans and guns) from other objects in a scene with sufficient detail. The saliency map is passed to the RCNN to recognize the human and gun. The recognition process results are shown in Fig. 9.

Figure 9 Detection results (in real time streaming.

(A) Detection of two humans in the sense; (B) sense output for humans detection; (C) detection of human with gun; (D) sense output for humans and gun detection.

As shown in Fig. 9, the CNN can successfully recognize humans in real time. The recognition had an accuracy of 99.2%, and it can detect an object with no slowdown or missing object failure. In particular, the system tracked multiple humans in only 0.9 s. The mean relative error was computed by normalizing the given values through the following formula: metric=mean|ypred−ytrue|∕normalizer

which is based on the TensorFlow metric “tf.keras.metrics.MeanRelativeError”, (TensorFlow, 2020). The mean relative errors of recognition are presented in Table 2.

Table 2 Recognition results.

Object	Error	
Human	4.3536820e−05	
Knife	2.6346240e−04	
Large Gun	9.1683286e−01	
Small Gun	8.2903586e−02	

By using transfer learning, we reduced the false positives, as indicated by the metrics. Specifically, the validation mask loss is approximately 0.475, and the validation class loss is approximately 0.0383.

We tested the system to recognize and detect a person in real working operations. The system first detects if there is any movement in the range of the ultrasonic sensor. If there is movement, the system then opens the camera and detects if there is a human in the field of view of the camera. If yes, then the system turns the light on and shows a screen and alarm to the surveillance officer. Figure 10 shows the detection process in a room.

Figure 10 Detection process of the proposed system.

(A) Human detection screen, a green square represents the detected human. (B) Human distance calculation (the system computes the distance of the human from a wall when entering the surveillance area and activates the light (light text in figure) at distance of 5 meter and making alarm with showing the camera view in monitor screen where human detect (TV text in figure).

As shown in Fig. 10, we tested the system’s operations when a human enters the surveillance area. When an ultrasonic sensor detects movement, it computes the distance and switches on the camera, thereby initiating the recognition process. If the system detects humans, the camera is continuously switched on to track the movement of the person and analyze the activity while continuously computing the distance. If the human enters a restricted area or has a weapon, the system alarms the observer and displays the stream from the camera. The system thus successfully detects humans and guns and isolates them from other objects in the area. The results indicate that the system successfully and effectively detects humans and guns with a high accuracy between 16% and 99% with a low response time of approximately 0.9 s. However, even in the case with low accuracy, detection and isolation are successfully performed for humans, even when a part of the human body is hidden. The algorithm in such cases exhibits reasonable performance for detection movement and computes distance with the error ranging between 0 and 5 m. Overall, the system can perform 100% detection for objects and can track humans and guns.

We also compared the recognition processes of the systems based on the PC and Raspberry Pi. The PC system has high computational hardware (core i7 8750 3.9 GHz CPU, 16 GB DDR4 RAM, a 256 GB NVMe SSD HDD, and an HD webcam). We compared the detection time and recognition percentage in a room with the same lighting conditions. The results are presented in Table 3 and Fig. 11.

Table 3 Recognition results for the systems-based PC and Raspberry Pi based System.

Number of persons in view	Detection Time for System Based on PC	Detection Time for System Based on Raspberry Pi	
	(s)	Accuracy (%)	(s)	Accuracy (%)	
1	0.69	99.4	0.71	99.1	
2	0.7	98.45	0.73	98.7	
3	0.7	96.4	0.79	97.8	
4	0.7	92.5	0.8	95.4	
5	0.73	93.1	0.86	94.3	
6	0.75	98.5	0.94	98.9	
7	0.74	99.4	0.98	98.7	
8	0.74	99.4	1.3	93.1	
9	0.75	99.4	1.98	91	

Figure 11 Results of detection for PC based systems and Raspberry Pi based System.

(A) The speed of the system response to the number of people in the scene. (B) Detection efficiency to the number of people in the scene.

Conclusions

In this work, we designed a smart surveillance system utilizing a low-cost computing unit and a CNN to monitor certain aspects and alarm observers automatically. A single board computer, a Raspberry Pi 3 version B, is used as the central controller that manages several tasks at the same time. For distance detection, a low-cost ultrasonic sensor type (HC-SR04) is used to sense motion and compute the distance from moving objects within the monitoring area. The recognition model can recognize and track the desired moving object (human) in real time, detect his/her activity (in this work, we focused on gun detection) and alarm officers if the situation is critical. The model is based on the faster RCNN optimized by using a saliency algorithm for feature extraction. Compared with the existing saliency methods, the proposed method does not require a database to identify objects, and it uses the local and global approach to generate a saliency map that enables fast and accurate feature extraction. The main results that have been achieved are described as follows:

• The overall system works smoothly and efficiently, and the controller can successfully control multiple tasks simultaneously with no failure.

• The recognition model operates promptly and accurately.

• The RCNN part of the proposed model is different from other surveillance approaches in that this model can use a low computational component, such as a Raspberry Pi, to perform multiple tasks with accurate and fast recognition. In this manner, a compact dedicated smart surveillance camera can be used to integrate this system to establish a control room to control a large number of surveillance cameras to perform multiple tasks, such as the surveillance of institutes, military bases, and cities.

• The recognition process is fast and highly accurate owing to the use of saliency algorithms with the RCNN. The main advantage of using a saliency algorithm is the reduction of the image details due to the removal of undesired features from the image and retaining only the critical objects in the scene. In this manner, the system successfully isolates the essential features and uses these features in the training/recognition process. The removal of unimportant objects along with a reduction in the image details can reduce the computational requirements and increase both the accuracy and speed of the training/recognition process.

• The most critical achievement of this work is the reduction in the computational requirements and improvement in the recognition process in both speed and accuracy. The results show that the system can recognize humans and threats (e.g., a human handling a gun) in any situation with recognition rates ranging from 16% to 99.4%.

• Even with a low recognition percentage, the system can successfully detect and classify humans and guns with an accuracy of nearly 100% in different situations (for instance, in cases in which a human is partially hidden behind certain objects).

• The model can work on low computational systems, such as a single board computer, with a fast processing time.

• The system achieved real-time detection of humans in less than 1 s when using both Raspberry Pi and PC models.

In summary, the model can perform fast recognition, which is essential in surveillance systems. However, the model needs more research and improvement, and I suggest the following:

• Use other architectures, such as the CNN, YOLO, SSD, mask RCNN, etc.

• Extend the system to detect other types of objects or even behaviors (e.g., theft or violence).

• Complement the system with another PC with more resources capable of performing online learning to retrain the system with new images.

Supplemental Information

Supplemental Information 1 Recognition Results

Click here for additional data file.

Supplemental Information 2 System Program

Click here for additional data file.

Supplemental Information 3 Run program video

Click here for additional data file.

Supplemental Information 4 Recognition result test

Click here for additional data file.

Additional Information and Declarations

Competing Interests

Author Contributions

Data Availability

The authors declare there are no competing interests.

Zaid Saeb Sabri conceived and designed the experiments, performed the experiments, analyzed the data, performed the computation work, prepared figures and/or tables, authored or reviewed drafts of the paper, and approved the final draft.

Zhiyong Li conceived and designed the experiments, authored or reviewed drafts of the paper, and approved the final draft.

The following information was supplied regarding data availability:

Raw data and code are available in the Supplemental File.

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
