# Peer review of "Low-cost intelligent surveillance system based on fast CNN"

_PeerJ Computer Science, doi:10.7717/peerj-cs.402_

## Round 0.1 · original submission · Major Revisions

Dear Authors,

In revising your manuscript please take particular attention to issues highlighted by second and third reviewers.

Moreover, please perform a carefully proofread of the entire manuscript.
before resubmission.

·

Basic reporting

• The word "low cost" should be updated or more defined in the Title article.
• Paper written in clear English language.
• The majority of the text is technically accurate.

==============================================
• There is no literature survey in this paper.

• This Paragraph need to be more clear.
”Line: 84 The efficiency of the existing surveillance systems depends on the effectiveness of human, “
“Line: 85 operators, who are required to monitor the camera footage.”
I suggest that you improve the description at lines 84 and 85 to provide more justification.
In general, Introduction is very good.

• There is Perfect and Enough theoretical background.
===============================================
• Paper in standard format as a structure.
• Some of Figures need to be more quality such as Figure 4, and Figure 8.
• Figure 9, I Think above title is incorrect. In addition, X-axis and Y-axis not clear.
• Dataset is clear and good for this track.
===============================================
• There are relevant results to all hypotheses in this paper.
• Most of the results mentioned in this paper are clearly defined.
• Line 274 “Software” this term need to be change I suggest “Implementation”
===============================================

Experimental design

• With most paper goals the results are described in detail.
• Rigorous work carried out to a high scientific & ethical level.
• Methods described with adequate detail & and no replicates are available.
• Line: 333 Step 2: Train samples: 2514. Val Sample: 486, I think this not clear.

Validity of the findings

• More clarification for the novelty in this paper is needed.
• Data on the results are appropriate and acceptable.
• The data is statistically accurate and reliable.
• The conclusions are well stated. However, I recommend being as points or bulleted list.

Additional comments

I would like to thank the researcher for his handling of this important topic and I think he addressed the topic in a very good theoretical and practical way.

·

Basic reporting

1. General concepts in computer vision are very well explained, like ReLU (lines 132-142) and pooling (lines 143-151). On the other hand, the explanation of saliency is hard to follow (lines 165-215). The scheme in figure 2 helps to understand saliency, though. Good job on including it.

2. Your references are sufficient and properly displayed at the end of the article. I suggest you cite them in the text. For instance, claims like the ones you make in lines 89-90 and 105-106 would greatly benefit from pointing out which reference backs them up.

3. Table 1 needs to specify what metric is following (relative error?) .

4. Figure 9 has an errata in its title: change “recognation” to “recognition”.

5. Please consider adding some of the following definitions – or cite a source that explains them: artificial immune system (line 110), the integration structure “o” (line 213), BMS and LG algorithms (lines 315-316), superpixels (line 317).

6. The image in figure 1 is watermarked. You should ensure you have the rights to use the image and credit the original source.

7. The word “realize” is repeated several times throughout the article, with the intended meaning of “perform” (i.e. lines 23, 31, 40). I recommend you avoid using “realize” in those context as its meaning can easily be missunderstood as “notice”.

Experimental design

1. The methods you describe are not enough to reproduce the CNN model. Please include the base model you used for transfer learning. You list some good ones as examples (line 303), but forget to specify which one(s) you used.

2. I strongly suggest you detail a bit more the training process: which layers you freezed, how many epochs you allowed the model to train, if you applied early stopping, the type of regularization used, etc.

3. I commend your idea of including the steps to follow in order to build the RCNN (lines 331-341). However, steps 3, 8, 9 and 10 does not pertain to the algorithm process. Rather, they are observations on the data and the model results.

Validity of the findings

To my knowledge, both the training, validation and test data came from the same source. Consider testing your model against images from other sources. Andrei Barbu et al. (ObjectNet: A large-scale bias-controlled dataset for pushing the limits of object recognition models) pointed out that computer vision models tend to learn the bias of a dataset when training (i.e. similar objects always being in front of a similar background). Showing your CNN gets good results against new datasets will prove it is robust and trustworthy.

Additional comments

I thank for providing the model and the script to run and test it. I must note, however, that your code looks very similar to the one that Adrian Rosebrock writes in his blog Pyimagesearch. If you have used his work for reference, I strongly encourage you to include Pyimagesearch between your references.

Reviewer 3 ·

Basic reporting

The paper is written in a good quality English and reads well. Minor revisions are necessary to make sure that the paper is self contained. Please remember that PeerJ is not a NN community journal. You should explain things like alpha coefficients (line 189) and CNN, RPN, ROI pooling (line 326). Make sure that all the terms and concepts used in the paper are explained.

The major problem of the paper has not described any related works and the text has no references in it. Before the paper is accepted the following two improvements must be done:
1) The existing related works must be analysed. Before doing your research, you must have defined the problem and checked the existing solutions. Maybe the problem is already solved and there is no need for additional research? I truly believe that it is not the case, but you have to prove it in the paper by showing that you know what the related works are and what are their shortcomings.
2) Appropriate referencing must be done in the text. Whenever some thoughts are coming from the literature, the reference must follow. All the papers that are in your reference list must be referenced from the text.

Minor things:
1) I believe Figure one contains copyrighted image that is identified by a watermark. Such image can not be used in your paper.
2) Quality of Figure 8 is low. Can it be improved somehow?
3) The used terminology must be improved to increase the accuracy of the way how the ideas are expressed. For example, line 32: is it about range of activations or range of activation functions? Line 275: is the system designed in Python or developed in Python?
4) Lines 331-341. Are these really steps? What is done in step 2? No description available, just some numbers. Also step 3. A single class includes fewer data. Similarly, Step 10.

The code is shared, but it is not well enough described. There is just a source with some comments in it. It is not clear, where can the reader find the most important things. This must be explained better.

There is also something wrong with figure numbering. Since the figures do not correspond to the captions.

Experimental design

The research fully complies with the scope of the journal. Still, the knowledge gap is not well defined since there is no description of related works and no references in the text.

The experiments seem sound and the results are of quite high accuracy, still the experimental setup is not well enough described. You should describe under what conditions the system was tested. Was it tested only on the images from validation data set or some kind of system's performance in at least laboratory environment was done?

Line 395: You are stating that the system has "high accuracy between 16% and 99%". Is 16% high accuracy? If so, then you should explain it in details. What is the accuracy of other systems?

The used data set is not provided (if it is in the supplemented files), it was not possible to check it without any further instructions available.

Validity of the findings

Only the final results are provided. it is unclear how exactly the experiment was done. Was there any real deployment in laboratory environment or only the object recognition part was done in the experiments? Figure 8 that should show the setup is missing.

Additional comments

The paper is interesting and shows a nice application of CNNs to partial automation of video surveillance systems. Technically the implementation seems good. Still, a journal paper still must be written by providing all the details about the experiments and positioning your research among all other efforts done within this area, where lately many works exist.

·

Basic reporting

The method exposed is well explained and easily followed with a minimal understanding of the Computer Vision field. The text is well written and understandable, even for a non-native English speaker. The article has a good structure that explains all the important and interesting points of the research. Figures and tables complement the text giving relevant information about the methods, process and the obtained results.

On the other hand, the article has a mayor problem that is the lack of references inside the text itself. This makes it difficult to to read the references at some point. For example:
- In page 3 line 105 "Several recognition approaches involving the support vector machine (SVM), artificial neural networks (NN), decision trees, and other rule-based classification systems have been proposed." in which some methods are described and it would be useful to check the linked articles/books/other.

- In page 3 line 116 "For instance, it is considered that the perpetrators often visit various places at different times before committing crimes, and they can thus be detected and located at different sites" its hard to find the reference in the Bibliography. The exact reference should be inserted here.

Other points to consider are:

- Figure 1 has a watermark. There could be problems with this figure, it is better to change it.

- Page 3 line 122 "fast R-CNN" should be corrected to "Faster R-CNN", it is a different method than the used here. This also occurs in line 419.

- The first paragraph of the section "Hardware Design" may be changed or complemented with a table with the costs of all of the components. Also, the price of the ultrasonic sensor is not indicated.

Experimental design

This article introduces a novel system to save costs without reducing the accuracy and increasing much the computational time of the method used (a Faster R-CNN). With this, it is intended to help in the task of indoor security, monitoring certain areas with surveillance cameras and doing the processing with various Raspberry Pi 3, having a central system in a low cost PC. The Neural Network architecture is well explained and the election of the layers are justified. However, the negative point of using pooling layers (loss of information) is not explained and should be mentioned in the section "System concepts".

Aditionally, transfer learning is used to take advantage of other already trained systems, saving the long training times for the selected architecture. Nevertheless, some references should be added to explain this term, such as the book "Deep Learning" by I. Goodfellow Et. Al. that has a specific chapter that explains it.

Also, other metrics could be used to improve the results section, such as precision and recall (and F1-score), specially for the dangerous objects. Moreover, the recall metric explains how many dangerous objects are correctly extracted and should be the metric to maximize.

As a minor comment, it would be useful to mention the Faster R-CNN architecture in the software section.

Validity of the findings

The authors don't present a novel model of identifying people and/or threats using security cameras nor a method for improving its speed. Instead, they present a complete system to perform this detection which is simple and cheap. With this, security systems could be complemented with the proposed one and also serves as a support for the security staff. Moreover, with the simplicity of the implementation and the reduced costs it could be widely adopted by different companies. Results present a sufficient way of performing the detection, being fast and accurate. Additionally, they present a way of reducing the processing costs, performing this operation only when a ultrasonic sensor detects movement.

As a comment, it is interesting to introduce a "Future work" section or paragraph, showing how this work can be extended, such as:

- Using other architectures for the CNN (YOLO, SSD, Mask R-CNN, etc.)

- Extend the system to detect other type of objects or even behaviours (like theft or violence).

- Complement the system with another PC with more resources capable of performing online learning to re-train the system with new images.

Additional comments

The research carried out by this article is interesting not only for the computer science community but for the general public, since it takes advantage of previous studies in the field of computer vision and uses it in the field of security, which is of general interest. On the other hand, mistakes in references should be corrected, which will improve the general reading, and change the figure with the watermark, which will avoid problems for the authors.

---

## Round 0.2 · Major Revisions

Please address the issues highlighted by the reviewers carefully, in particular the Reviewer 2 which highlighted relevant issues.

·

Basic reporting

First of all, goob job on taking into account all the relevant feedback from the previous review. The paper has improved substantially. However, I must say that in this document the English is sometimes lacking, specially when using verb tenses (i.e. the sentence on lines 73-74; "utilizes" on line 75). These makes the paper hard to read, even to the point that some sentences cannot be understood properly (i.e. lines 151-153). I strongly advice you to revise the paper carefully.
On the other hand, I commend your effort on properly displaying the references throughout the article.

Experimental design

All the experiment are now better explained and easier to follow and reproduce.  Some terms still lack definition, like "context-based propagation" on line 384, or "N order reachable matrix" on line 382. Defining these may be out of the scope of the paper, but adding a reference to an article explaining them would be welcomed.

Validity of the findings

I commend your effort on generating your own image dataset. If every picture was took by the same group - as the ones shown in the examples - I deeply encourage you to test your model against images from other sources - specially with other people appearing holding weapons and such. This would be very insightful and show whether the model is robust.
On line 471, when you say "In particular, the system tracked multiple humans in only 0.9 s.", is that a particular instance, or the mean tracking time? The first one would be very informative, while the latter isn't.

Additional comments

I am obligued to remark that your code looks very similar to the one that Adrian Rosebrock writes in his blog Pyimagesearch. Specially this post have identical lines: https://www.pyimagesearch.com/2019/04/15/live-video-streaming-over-network-with-opencv-and-imagezmq/. If you have used his work for reference, I strongly encourage you to include Pyimagesearch between your references.

Reviewer 3 ·

Basic reporting

The paper is written in a good quality English and reads well. Still it should be reread for minor typos. Some examples that must be corrected:

Line 126: resurches
Line 222: where each of it is pixel's shows
Line 422: second is to defined analysis
Line 426: is would be an anchor
Line 540: to achieved
Line 568: use of brackets.

The bibliography and use of references are sufficient. The state of the art is described adequately.

Terms mostly are explained for a reader that is familiar with the machine learning and neural networks. One minor remark: on line 476 explain the essence of the metrics used instead of providing a class name.

Experimental design

The research matches the scope of the journal and the explanation of the knowledge gap has been added.

The experiments seem sound and the results are of quite high accuracy, the description of the experimental setup has been detailed in the resubmitted version. You also described the accuracy of 16% in the rebuttal letter. I still believe that this can be better explained in the article. Consider this as reviewer's personal opinion.

Validity of the findings

The figure with experimental setup was added. The same as the data collection methodology and experimental setup description.

Additional comments

No further comments. I believe, the paper is ready to publish after minor technical improvements.

---

## Round 0.3 · Minor Revisions

Please in revising your manuscript take into account reviewer comments and perform careful proofreading of the entire work.

Reviewer 3 ·

Basic reporting

My main comments have been considered and mistakes have been corrected. A few new mistakes regarding number agreement have been introduced, for example on line 551 "a multiple tasks". (You can not use "a" with plural), "every pixels of the image have" on line 222. Please carefully reread the whole paper to check English.

Experimental design

The only Major question is regarding the sentence: "However, I utilized works done by (Rosebrock A., 2019) as a reference for design CNN model." If you have used some other person's work as a basis of your own solution, then you have to clearly show what is your contribution - what has been improved from the original code? What were the problems in the original solution? Why did it need any improvements? Does your solution solve the identified problems? I mean, you have to clearly position your work within the current state of the art and clearly show your contribution.

Validity of the findings

The clarification regarding the abovementioned use of other person's works must be done to make sure that the novelty of the paper is significant enough.

Additional comments

No further comments.

---

## Round 0.4 · accepted · Accept

Please perform a final proofreading to improve English quality. There are still some mistakes and the writing could be improved.